# Effects of *Trichoderma harzianum* on Growth and Rhizosphere Microbial Community of Continuous Cropping *Lagenaria siceraria*

**DOI:** 10.3390/microorganisms12101987

**Published:** 2024-09-30

**Authors:** Jinlei Wang, Hongmei Mu, Shan Liu, Saike Qi, Saifeng Mou

**Affiliations:** College of Agriculture and Biology, Liaocheng University, Liaocheng 252000, China; 15095270742@163.com (J.W.); 15216360457@163.com (S.L.); 13678637432@163.com (S.Q.); saifengmu01@163.com (S.M.)

**Keywords:** chemical properties, rhizosphere soil, soil enzyme activity

## Abstract

This study analyzed the effects of *Trichoderma harzianum* on the growth of continuous cropping *Lagenaria siceraria* and the physical and chemical properties of rhizosphere soil and microbial community structure, using Illumina Miseq (PE300) high-throughput sequencing technology along with physiological and biochemical detection. The results indicated that after applying *T. harzianum*, the growth of *L. siceraria* was significantly promoted, with increases in plant height, fresh weight, and dry weight of 21.42%, 24.5%, and 4.5%, respectively. The pH of the rhizosphere soil decreased from 7.78 to 7.51, while the electrical conductivity, the available phosphorus, the available potassium, and the total nitrogen were markedly higher compared to the control group and increased by 13.95%, 22.54%, 21.37%, and 16.41%, respectively. The activities of catalase and sucrase in the rhizosphere increased by 18.33% and 61.47%, and the content of soil organic carbon (SOC) increased by 27.39%, which indicated that *T. harzianum* could enhance soil enzyme activity and promotes the transformation of organic matter. The relative abundance of beneficial bacteria such as *Pseudomonas* increased, while the relative abundance of harmful fungi such as *Fusarium* and *Podosphaera* decreased significantly.

## 1. Introduction

*Lagenaria siceraria* (Molina) Standl, commonly known as* L. siceraria*, gourd, or lupa, is an annual climbing vine herb [1]. *L. siceraria* possesses substantial nutritional and economic value due to its chemical composition, which includes glucose, pentosan, lignans, and other compounds [2]. These components contribute to its diuretic and anti-inflammatory effects, thereby enhancing its medicinal value [3]. The annual sales of *L. siceraria* crafts in Liaocheng, Shandong Province, China reached more than a few million dollars. In recent years, the intensive planting and improper field management measures of *L. siceraria* have caused the emergence of continuous cropping barriers. This induced the quality and yield of *L. siceraria* fruit to continue to decline, which has emerged as the primary obstacle to the advancement of the *L. siceraria* industry.

Continuous cropping refers to a planting model in which a single crop is cultivated on the same land for an extended period. This planting model is prone to continuous cropping obstacles, which are manifested by growth impediments, exacerbation of pests and diseases, yield reduction, and quality decline [4,5,6]. The resistance and stability of soil microorganisms within the soil ecosystem, as well as their roles in the circulation of soil materials, are critical. These microorganisms facilitate water transport, organic matter decomposition, nutrient absorption, and soil transformation, significantly impacting plant disease and pest management [7]. After long-term continuous cropping of watermelon, the spectrum of rhizosphere bacteria decreased, whereas the spectrum of fungi increased, with a corresponding increase in the relative abundance of pathogenic fungi like *Fusarium*, which became the leading dominant group [8]. Similarly, continuous cropping of cucumber significantly increased the amount of fungi in rhizosphere soil, with a notable rise in harmful *Fusarium* species and a decline in the prevalence of beneficial bacteria such as *Pseudomonas* [9]. Long-term continuous cropping of Cucurbitaceae plants maybe lead to alterations in the number, biodiversity, and species richness of soil rhizosphere microorganisms, and structural imbalance is a significant factor related to the development of continuous cropping challenges [10]. Therefore, addressing these obstacles by modifying the soil rhizosphere microbial population structure is viewed as an effective strategy.

*Trichoderma harzianum* is a biological control agent that effectively alleviates the sustained cropping obstacles in the production of cucumber, melon and pepper production [11,12]. It not only inhibits the survival and activity of pathogens, preventing soil pathogens multiplication, but also enhances plant growth, increases resistance to stress and disease, and comprehensively improves the quality of crops [13,14]. *T. ghanense* and *T. citrinoviride* have been shown to restrict the growth of cucumber pathogens such as *Pythium aphanidermatum* while facilitating cucumber growth [12]. Following the application of *T. saturnisporum*, the damage caused by *Pythium ultimum* to melon seedlings was reduced, with disease severity decreasing by 63% [15]. In greenhouse experiments, the execution of *T. harzianum* effectively curtailed the incidence of peanut brown root rot and promoted plant growth [16]. The concurrent application of biochar and *Trichoderma* increased the variability of bacterial communities within the rhizosphere soil of continuous cropped cucumber, optimized the physical and chemical attributes of the soil, and stimulated root activity [17]. Under sterile laboratory conditions, the application of *Trichoderma harzianum T-22* biofertilizer to the *carrot* rhizosphere soil significantly increased the numbers of *Bacillus* and *Pseudomonas*, as well as beneficial fungi such as *Trichoderma* [18]. Additionally, inoculation of *T. harzianum* ST02 in the rhizosphere soil of sweet sorghum significantly increased the relative abundance of beneficial nitrogen-fixing bacteria, such as *Arthrobacter* [19].

Therefore, the effects of *T. harzianum* were evaluated on the performance of *L. siceraria*, as well as on the physico-chemical properties and microbial community structure of the rhizosphere soil under continuous planting conditions. This study aimed to evaluate the remediation potential of *T. harzianum* for continuous *L. siceraria* cultivation and to provide a scientific basis for effectively alleviating the adverse effects of continuous cropping.

## 2. Materials and Methods

### 2.1. Experimental Materials

*T. harzianum* (T-22) was produced by Shandong Cangyuan Biotechnology Co., Ltd. (Weifang, China) (effective viable count: ≥1 billion/g).

The experimental material *L. siceraria* variety is Yayao gourd, which is the largest gourd variety planted in Liaocheng, Shandong Province. It is preserved by the Muhongmei Research Group of the College of Agriculture and Biology of Liaocheng University.

### 2.2. L. siceraria Cultivation Methods

The soil used for the pot experiment was sourced from the plantation of the College of Agronomy and Biology at Liaocheng University, located in Liaocheng City, Shandong Province, China, where *L. siceraria* has been continuously cultivated for two years. This region falls within the temperate monsoon climate zone, with an annual average temperature of approximately 13.1 °C, annual precipitation of about 600 mm, and annual frost-free period of about 200 days. Before planting, the soil had a pH of 7.23, organic carbon content of 8.20 g/kg, total nitrogen of 710.45 mg/kg, available phosphorus of 45.63 mg/kg, available potassium of 121.52 mg/kg, and electrical conductivity of 0.56 mS/cm.

The *L. siceraria* was planted in a black plastic basin (bottom length 6.8 cm, height 8.5 cm, top length 10 cm), which was filled with *L. siceraria* rhizosphere soil (500 g) that had been continuously planted in the same field for two years. The crop continues (CC) group was irrigated with sterile water, and the crop treatment (CT) group was irrigated with 200 times diluted fertilizer per kilogram of liquid. Each treatment had 5 replicates and was irrigated every 5 days. The conditions of the greenhouse were as follows: room temperature 25 °C, relative air humidity 50%. Other growth conditions remained unchanged when growing in the controlled environment chamber of the College of Agronomy and Biology of Liaocheng University for 25 days, growing to the climbing stage.

### 2.3. Soil Sample Collection

After extracting the soil from the pot of *L. siceraria*, the soil was softly tapped away from the root system, and the soil sample adhered to the root exterior was coated utilizing a sterile brush. Then, the plant debris and root residues were removed through a 40-mesh sieve. The rhizosphere soil samples from the same pot were thoroughly combined to achieve a completely uniform sample. Frozen at −80 °C for storage, the soil microbial sequencing was entrusted to Shanghai Meiji Biological Company (Shanghai, China) (https://www.majorbio.com/, accessed on 15 July 2024) to analyze the chemical composition and enzymatic functions of the remaining soil.

### 2.4. Detection of Soil Chemical Properties

The soil pH and electrical conductivity were measured by potentiometric method and conductivity method (with a water-to-soil ratio of 2.5:1). The contents of soil organic matter, total nitrogen, available phosphorus, and available potassium were quantified by the potassium dichromate volumetric method, Kjeldahl method, combined extraction–colorimetric method, and flame photometer method, respectively [20,21,22].

### 2.5. Examination of Soil Enzyme Activity

Soil urease activity was measured using the indigo colorimetric method [23], soil sucrase activity was assessed by the 3,5-dinitrosalicylic acid colorimetric method [24], neutral phosphatase activity in the soil was measured using the disodium phenyl phosphate colorimetric method [23], and soil catalase activity was evaluated through potassium permanganate titration [25].

### 2.6. Soil Microbial Sequencing

Microbial genomic DNA was extracted from 0.25 g soil samples according to the E.Z.N.A.^®^ Soil DNA Kit (Omega Bio-tek, Norcross, GA, USA). DNA quality and concentration were assessed via 1.0% agarose gel electrophoresis and the NanoDrop2000 spectrophotometer (Thermo Scientific, Waltham, MA, USA). The hypervariable regions V3–V4 of the bacterial 16S rRNA gene were amplified with primer pairs 338F (5′-ACTCCTACGGGAGGCAGCAG-3′) and 806R (5′-GGACTACHVGGGTWTCTAAT-3′), and the ITS1 rDNA regions of fungi were amplified with a forward primer ITS1F (CTTGGTCATTTAGAGGAAGTAA) and reverse primer ITS2R (GCTGCGTTCTTCATCGATGC) pair, using a T100 Thermal Cycler PCR thermocycler (BIO-RAD, Hercules, CA, USA) [26]. PCR products were purified from a 2% agarose gel using the PCR Clean-Up Kit (YuHua, Shanghai, China), and quantified with a Qubit 4.0 fluorometer (Thermo Fisher Scientific, Waltham, MA, USA).

### 2.7. Sequencing Data Processing

The soil microbial diversity was sequenced using the Illumina Miseq PE300 platform (Illumina, San Diego, CA, USA). The purified amplicons were initially assembled according to the standard of equimolar amounts, and then subjected to paired-end sequencing.

The raw FASTQ files were de-multiplexed and then quality filtered using fastp version 0.19.6 [27] and merged to a reference standard using FLASH version 1.2.7 [28]. The representative sequences of bacteria and fungi were classified [29,30] and analyzed separately using an RDP Classifier version 2.2 [31] with a confidence level of 70%.

### 2.8. Data Statistics and Analysis

Using Mothur version 1.30.2, the dilution curve and α-diversity index were calculated. The Bray–Curtis distance matrix served as the basis for Principal Coordinate Analysis (PCoA) (Vegan v2.5-3 package). Species differences were assessed using Linear Discriminant Analysis Effect Size (LEfSe) (http://galaxy.biobakery.org/, accessed on 15 July 2024) software, with a significance level set at* p* < 0.05 and an LDA effect size threshold of >3.5. FUNGuild software (http://www.funguild.org/, accessed on 15 July 2024) and the Phylogenetic Investigation of Communities by Reconstruction of Unobserved States (PICRUSt, version 1.1.0) were employed to obtain functional predictions of bacterial and fungal communities. The Duncan’s Multiple Range Test was used to examine differences across treatments, with significance levels set at *p* < 0.05 and *p* < 0.01. All results are presented as mean ± standard deviation (SD).

## 3. Results

### 3.1. Effects of T. harzianum on Growth Indexes of L. siceraria

The application of *T. harzianum* was found to enhance the growth of *L. siceraria*. Specifically, this treatment resulted in a significant increase in plant height, as well as both fresh and dry biomass (Table 1). The height of the plants, along with their fresh and dry weight, exhibited increases of 21.42%, 24.5%, and 4.5%, respectively, compared with the CC group.

### 3.2. Effects of T. harzianum on Soil Physical and Chemical Properties of L. siceraria

The application of *T. harzianum* notably altered the chemical characteristics of the rhizosphere soil (Table 2). In the CT group, the electrical conductivity, the organic matter, the available phosphorus, the available potassium, and the total nitrogen were markedly higher compared to the CC group and increased by 13.95%, 27.39%, 22.54%, 21.37%, and 16.41%, respectively. The pH of rhizosphere soil decreased from 7.78 to 7.51. This suggests that *T. harzianum* enhances soil nutrient levels while also affecting soil pH.

### 3.3. Effect of T. harzianum on Enzyme Activity in Rhizosphere Soil of L. siceraria

The application of *T. harzianum* preparations enhanced the activities of urease, phosphatase, catalase, and sucrase to different extents (Table 3). The soil treated with *T. harzianum* exhibited substantial increases in catalase and sucrase activities, with enhancements of 18.33% and 61.47%, respectively, compared to the CC group.

### 3.4. Effects of T. harzianum on Soil Microbial Diversity and Species Richness

#### 3.4.1. Microbial Cluster Analysis

Following the quality control and filtering of sequencing data, a total of 56,149 and 53,216 bacterial 16S rRNA sequences, and 77,010 and 66,073 fungal ITS sequences were obtained from the CC and CT groups, respectively. Sequences exhibiting over 97% similarity to OTU representative sequences were chosen for annotation and further analysis. The Sobs diversity curves for both bacteria and fungi did not reach a plateau at a distance of 0.03 (Figure 1a,b), suggesting that the sequencing data may not fully capture all the communities present in the samples. In contrast, the Shannon diversity curves approached a plateau as the number of reads increased (Figure 1c,d), indicating that the sequencing data adequately represent the majority of the microbial diversity within the samples.

A similarity threshold of 97% was applied and a Venn diagram was utilized to depict the number of unique and shared OTUs across various treatments (Figure 2). The analysis revealed 3611 common bacterial OTUs between the treatments. The CC group exhibited 1560 unique OTUs, while the CT group had 1615. Additionally, 519 fungal OTUs were common, with 283 unique to the CC group and 322 unique to the CT group.

#### 3.4.2. Microbial Alpha and Beta Diversity Analysis

Notable differences were detected in the Shannon and Simpson indices in the fungal communities between the CC and CT groups (Table 4). Specifically, the Shannon index was higher in the CT group compared to the CC group, whereas the Simpson index was lower in the CT group. It is important to highlight that a higher Shannon index indicates greater community richness, while a lower Simpson index reflects increased community evenness. The indices for bacterial community diversity and richness did not differ significantly between the CC and CT groups. Similar results were observed for the Chao1 and ACE indices in the fungal communities.

PCoA ordination was conducted on various groups based on the OTU data (Figure 3). The analysis revealed that the application of *T. harzianum* notably altered the microbial community structure in the rhizosphere soil of *L. siceraria*. For bacteria, the PCoA (Figure 3a) indicated that the first and second principal components accounted for 45.23% and 21.88% of the total variance, respectively. In the case of fungi, the first and second principal components explained 62.54% and 13.79% of the variance, respectively.

### 3.5. The Community Composition and Structure of CC and CT Groups

Following sequence classification with the Mothur program, the sample data were annotated and summarized. At the phylum level (Figure 4a and Figure 5a), the predominant phyla in the soil bacterial community were Proteobacteria, Actinobacteriota, Chloroflexi, Acidobacteriota, and Bacteroidota. These phyla constituted 75.49% of the bacterial community in the CC group and 79.22% in the CT group. Among them, Proteobacteria had the highest proportion in the two groups, with 27.39% and 25.19% for the CC and CT groups, respectively. Among the top 12 most abundant bacterial phyla, Bacteroidota, Patescibacteria, and Armatimonadota were found at significantly higher levels in the CT group compared to the CC group (*p* < 0.05). Additionally, Deinococcota showed a significantly greater presence in the CT group (*p* < 0.01). Conversely, Myxococcota and Bdellovibrionota were significantly less abundant in the CT group relative to the CC group. Overall, the distribution of soil bacteria in the two groups was similar at the phylum level. At the phylum level, the soil fungi in the samples were analyzed and annotated (Figure 4c and Figure 5c). The predominant fungal phyla were Ascomycota, Mortierellomycota, and Basidiomycota, which together comprised 86.9% and 91.39% of the fungal community in the CC and CT groups, respectively. Notably, the relative abundance of Ascomycota far exceeded the proportion of other phylums, which was 77.18% and 77.12% in the CC and CT groups, respectively, and was the most important component of the entire fungal community. The relative abundance of Basidiomycota was significantly higher in the CT group compared to the CC group (*p* < 0.001), while Glomeromycota was found at significantly lower levels in the CT group.

At the genus level, the bacterial differences in the data of different treated samples were analyzed (Figure 4b and Figure 5b). Significant differences in genus distribution were observed. After the application of *T. harzianum*, *Pseudomonas* displayed a markedly higher relative abundance compared to the continuous cropping (CC) group. In contrast, the levels of *Microvirga*, *Haliangium*, and *Ilumatobacter* were notably reduced in the *T. harzianum*-treated samples relative to the CC group. Similarly, analysis of the fungal community at the genus level (Figure 4d and Figure 5d) revealed substantial differences in fungal genus distribution across treatments. *Penicillium*, *Podosphaera*, and *Talaromyces* were significantly more abundant in the CC group compared to the CT group (*p* < 0.05). The abundance of *Preussia* was significantly higher in the CC group (*p* < 0.01). Conversely, *Conocybe* was more prevalent in the CT group (*p* < 0.05), and *Mycothermus* was also significantly more abundant in the CT group (*p* < 0.01). Additionally, *Lophotrichus*, *Trichosporon*, and *Fusicolla* were present at significantly higher levels in the CT group compared to the CC group (*p* < 0.001). Furthermore, *Fusarium* showed a higher proportion in the CC group, with relative abundances of 8.86% and 8.21% in the CC and CT groups, respectively. Previous studies have shown that *Fusarium* and *Podosphaera* can cause a variety of diseases in plants.

### 3.6. Microbial LEfSe Analysis

Linear discriminant analysis (LEfSe) was employed to assess differences in bacterial and fungal communities across four treatment groups, aiming to identify microbial species with significant variations in relative abundance. Species differentiation was determined using an LDA score greater than 3.5. The results of bacterial community analysis (Figure 6a and Figure 7a) showed that the biomarkers in the CC group included the following: *Microvirga*, *Cyanobacteriia*, *Polyangia*, *Chloroplast*, *Polyangiales*, and *Cyanobacteriales*. The biomarkers in the CT group, such as Pseudomonas, Flavobacteriaceae, and Streptosporangiales, were significantly enriched (*p* < 0.05). Additionally, these biomarkers demonstrated substantial enrichment (*p* < 0.05) within this group. The results of fungal community analysis (Figure 6b and Figure 7b) showed that the biomarkers in the CC group included the following: *Preussia*,* Penicillium*, *Funneliformis*, *Cladorrhinum*, *Furcasterigmium*, *Thyridium*, *Podosphaera*, and *Myrmecridium*, and they were significantly enriched (*p* < 0.05). The biomarkers in the CT group mainly include the following: *Lophotrichus*, *Mycothermus*, *Trichosporon*, *Fusicolla*, *Albifimbria*, *Conocybe*, *Coprinellus*, *Trichurus*, *Rhodotorula*, *Hormiactis*, and *Thermomyces*, and were significantly enriched (*p* < 0.05).

### 3.7. Functions of Bacterial and Fungal Communities

Functional prediction analyses provide an initial interpretation of the microbial community functions within the samples (Figure 8). The application of *T. harzianum* significantly affects the relative abundance of soil fungal communities (Figure 8b). Among the dominant fungi with higher abundance, the relative abundance of Animal Pathogen-Endophyte-Fungal Parasite-Plant Pathogen-Wood Saprotroph, Animal Pathogen-Endophyte-Lichen Parasite-Plant Pathogen-Soil Saprotroph-Wood Saprotroph, Dung Saprotroph-Plant Saprotroph, and Plant Pathogen were significantly higher in the CC group compared to the CT group. Conversely, in the CT group, the relative abundance of Endophyte-Litter Saprotroph-Soil Saprotroph-Undefined Saprotroph fungi was notably higher than in the CC group. The primary functions of the dominant bacterial communities include (Figure 8a) the following: amino acid transport and metabolism, translation, ribosomal structure and biogenesis, energy production and conversion, and biosynthesis of cell wall/membrane/envelope components. No significant differences were observed between the CT and CC groups in these functions (Figure 8a).

## 4. Discussion

### 4.1. The Promoting Effect of T. harzianum on the Growth of L. siceraria

Dry weight and fresh weight are important bases for evaluating plant adaptability and stress resistance [32]. *Trichoderma*, when used as a biofertilizer, can promote plant growth and induce plant disease resistance. Research has indicated that under greenhouse and field conditions, the application of irrigation methods using the *Trichoderma strain TH1* can enhance the dry weight of *lentil plants* by as much as 120% [33]. *T. harzianum* has a beneficial effect on the fresh weight and moisture content of *Cannabis sativa* inflorescences [34]. In addition, *T. harzianum* significantly promoted the stem length and whole plant weight of *Bupleurum chinense* [35]. In pot experiments, mixing *T. harzianum* with soil from continuously cropped apple significantly promoted the growth of apple seedlings and increased enzyme activity in the rhizosphere soil [36]. Under greenhouse conditions, spraying *T. harzianum* TW21990 liquid on strawberry leaves also significantly enhanced strawberry growth and development. These results indicate that *T. harzianum* can promote plant growth through various application methods [37]. In this study, irrigation was used to treat continuously cropped *L. siceraria* seedlings, resulting in significant increases in plant height, dry weight, and fresh weight, similar to previously reported effects of other types of fertilizers on crop growth.

### 4.2. Effects of T. harzianum on Soil Chemical Properties and Enzyme Activities

Soil pH is a crucial indicator of soil health, and too acidic and alkaline soil environments are not conducive to plant growth [38]. In this study, the pH value was reduced from 7.78 to 7.51, which was more conducive to the survival of soil microorganisms suitable for the growth of gourd. Previous studies have demonstrated that functional microbial fertilizers can substantially enhance the availability of soil nutrients and alter the composition of soil microbial communities. This, in turn, regulates the accessibility and supply of rhizosphere nutrients, thereby improving soil fertility [39]. For instance, inoculating the rhizosphere of sweet sorghum with *Trichoderma* under salt stress led to significant increases in TN, AP, AK, and organic matter [19]. In this study, the changes in soil after applying *T. harzianum* are consistent with previous research. Soil enzymes, originating from microorganisms, plants, and animals are essential for the metabolic processes within soil ecosystems and play a vital role in nutrient cycling and energy flow [39]. The combined application of biochar and *Trichoderma* not only enhanced soil electrical conductivity, organic matter, AP, AK, and TN but also significantly boosted the activities of soil catalase and invertase [9]. Similarly, treatment with *T. harzianum* MHT114 in continuously cropped pepper fields resulted in increased levels of TN, organic matter, AP, and AK, alongside heightened soil enzyme activity [11]. In this study, the activities of urease, phosphatase, catalase, and sucrase in the soil were significantly increased after the application of *T. harzianum*, which was consistent with the reported studies. *T. harzianum* increased the activity of soil enzymes, promoting the transformation of nutrients, and improved soil conditions.

### 4.3. Effects of T. harzianum on Microbial Community Structure in the Rhizosphere Soil of L. siceraria

Proteobacteria dominated in different treatments. Proteobacteria, being the most prevalent in soil, plays a critical role in the carbon cycle. Following the application of *T. harzianum*, the relative abundance of Bacteroidota, Patescibacteria, and Armatimonadota in the CT group notably exceeded that observed in the CC group. These phyla are commonly found in soils and are integral to nutrient transformation processes. Research has demonstrated that Bacteroidota can promote the absorption of N, P, K, and other nutrients in plants, and can improve the defense stress ability of plants [40]. Proteobacteria can increase the content of soluble phosphorus in soil and the utilization rate of phosphorus fertilizer by plants to promote plant growth [41]. Armatimonadota can degrade complex carbohydrates [42]. This is consistent with previous studies. After the application of rhizosphere probiotic *Bacillus*, plant growth was promoted, and Proteobacteria was also induced to a significant enrichment of Proteobacteria in the plant rhizosphere [43]. In addition, many strains of *Pseudomonas* are plant growth-promoting rhizobacteria [41]. It was confirmed by Lefse analysis that it was a significantly enriched biomarker in the CT group. Similarly, inoculation of soil with *Trichoderma* M45a increased the relative abundance of *Pseudomonas* in the rhizosphere and promoted plant growth, corroborating the results of this study [44]. Thus, the application of *T. harzianum* markedly enhances the relative abundance of beneficial bacteria, such as Pseudomonas, within the bacterial community.

Ascomycota represents the predominant phylum within fungal communities and constitutes a major element of the overall fungal population. As the largest and most diverse fungal phylum in soil, it is a key organic decomposer. *Fusarium* can survive in normal plants and can cause plant disease when conditions are suitable. Previous studies have shown that *Fusarium* causes replant disease of apple trees [45]. After continuous cropping of watermelon, the concentration of *Fusarium* spores in the soil increased, which enhanced its reproductive capacity [46]. In this study, the application of *T. harzianum* resulted in a reduced proportion of *Fusarium* and *Podosphaera* in the CT group compared to CC group. At the same time, LEFse analysis confirmed that *Podosphaera* was a significantly enriched biomarker in the fungal community. This observation aligns with previous findings. The addition of *Bacillus subtilis* significantly reduced the abundance of *Fusarium* in the rhizosphere soil of *Panax notoginseng*. The application of *Trichoderma asperellum* also significantly reduced the proportion of *Fusarium* [47]. Thus, the incorporation of *T. harzianum* enhances plant growth by modifying the rhizosphere microbial community structure, increasing the relative abundance of beneficial bacteria like *Pseudomonas* and decreasing the prevalence of *Fusarium* and *Podosphaera*.

### 4.4. Effects of T. harzianum on Microbial Community Function in the Rhizosphere Soil of L. siceraria

Functional prediction of fungi revealed significant differences between the CC and CT groups. Previous studies have indicated that an increase in Animal Pathogen-Endophyte-Lichen Parasite-Plant Pathogen-Soil Saprotroph-Wood Saprotroph and a decrease in Endophyte-Litter Saprotroph-Soil Saprotroph-Undefined Saprotroph are associated with the onset of saffron rot compared to healthy saffron [48]. Similarly, research on Perilla frutescens showed that an increase in Plant Pathogen within the microbial community led to a decline in soil nutrients and inhibited plant growth [49]. These findings align with our results, where the relative abundance of harmful fungi was higher and beneficial fungi lower in the CC group compared to the CT group. In terms of bacterial functional prediction, no significant differences were observed between the CC and CT groups. This lack of difference might be due to the large diversity and abundance of soil microorganisms, where the functional changes in a few specific bacteria may not be reflected in the overall data. The application of *T. harzianum* was effective in reducing the abundance of harmful fungi and increasing the relative abundance of beneficial fungi, thereby improving the structure of the soil microbial community.

## 5. Conclusions

Using root irrigation methods with *T. harzianum* on soil from continuously cropped *L. siceraria* significantly promoted *L. siceraria* growth, improved the physico-chemical properties of the rhizosphere soil, and enhanced soil enzyme activity, thereby increasing the soil’s nutrient transformation capacity. High-throughput sequencing revealed that *T. harzianum* altered the microbial community structure of the soil, increasing the relative abundance of beneficial bacteria such as *Proteobacteria* while reducing the abundance of harmful fungi like *Fusarium* and *Podosphaera*. Therefore, *T. harzianum* is an effective microbial agent for alleviating the adverse effects of continuous *L. siceraria* cropping.

## Figures and Tables

**Figure 1 microorganisms-12-01987-f001:**
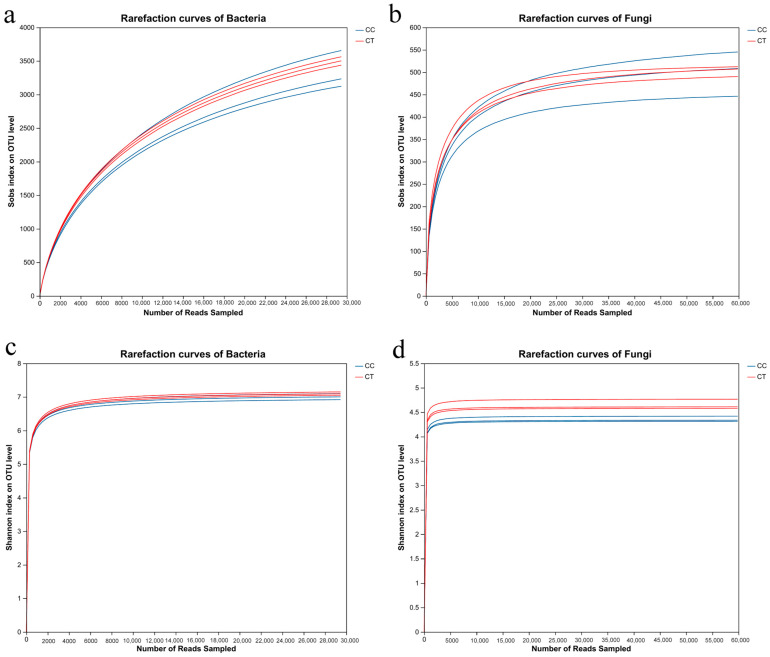
Dilution curves under CC and CT treatments. (**a**,**b**) represent the Sobs curves of bacteria and fungi, respectively, and (**c**,**d**) represent the Shannon curves of bacteria and fungi, respectively. CC refers to the continuous cropping treatment group; CT refers to the *T harzianum* treatment group.

**Figure 2 microorganisms-12-01987-f002:**

Unique and shared genera of the CC and CT groups in Venn diagram form. (**a**,**b**) represent bacteria and fungi, respectively.

**Figure 3 microorganisms-12-01987-f003:**
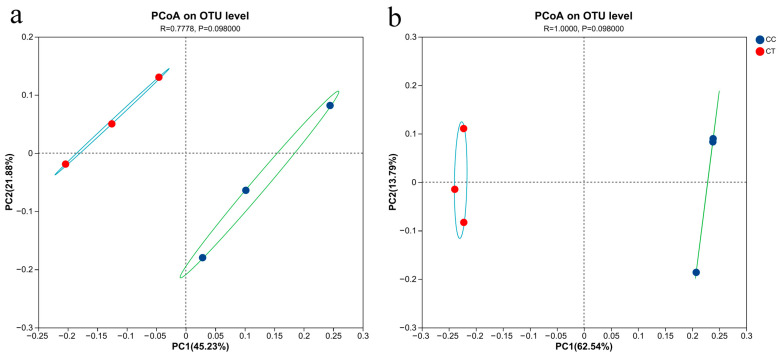
PCoA analysis under CC and CT treatments. (**a**,**b**) represent the β-diversity of bacteria and fungi, respectively. The blue dots represent the fertilizer treatment group of *T. harzianum*, and the red dots represent the continuous cropping treatment group.

**Figure 4 microorganisms-12-01987-f004:**
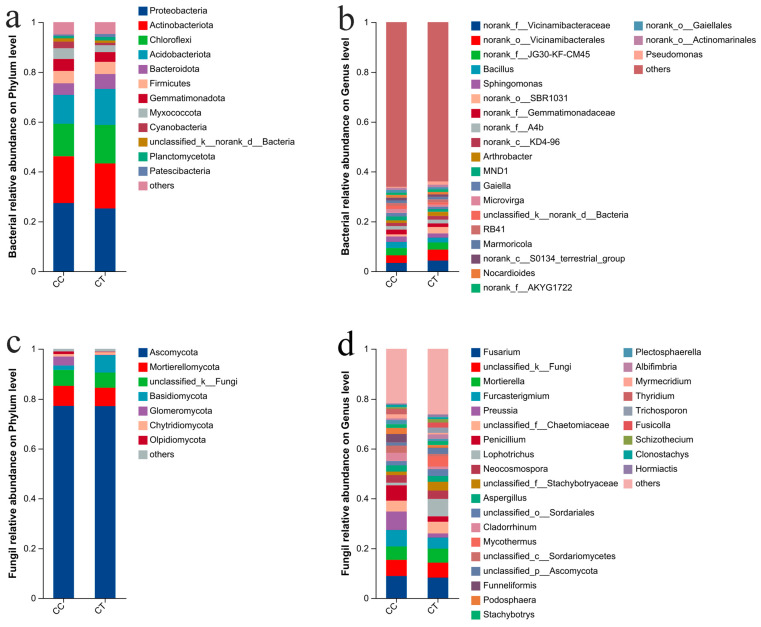
Composition analysis under CC and CT treatments. (**a**) Relative abundance of bacteria at the phylum level; (**b**) relative abundance of bacteria at the genus level; (**c**) relative abundance of fungi at the phylum level; and (**d**) relative abundance of fungi at the genus level.

**Figure 5 microorganisms-12-01987-f005:**
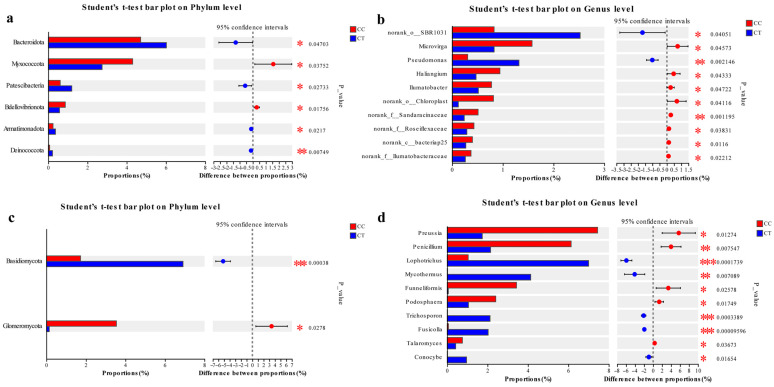
Difference analysis under CC and CT treatments. (**a**,**b**) are differences in bacterial communities at the phylum and genus levels. (**c**,**d**) are the differences in fungal communities at the phylum and genus levels. On the right side is the *p* value. * indicates that there is a significant difference between the two sets of data (*p* < 0.05), ** indicates that there is a very significant difference between the two sets of data (*p* < 0.01), *** indicates that there is a highly significant difference between the two sets of data (*p* < 0.001), using the Student’s *t*-test.

**Figure 6 microorganisms-12-01987-f006:**
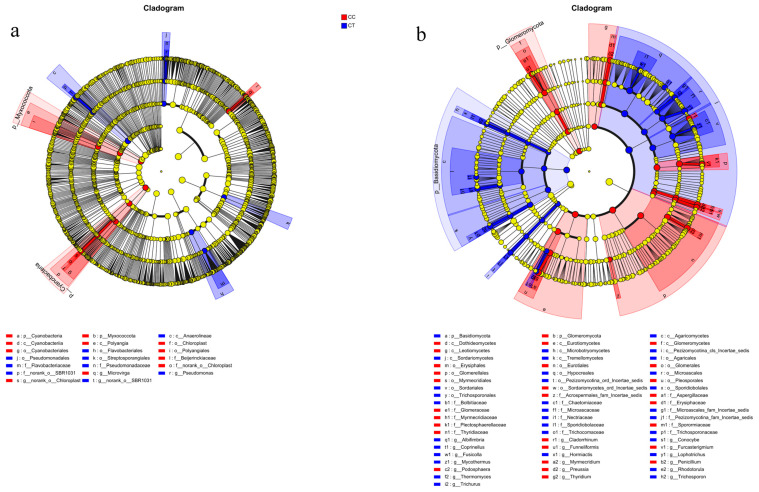
Difference analysis of community LEfSe under CC and CT treatments. (**a**) Bacteria. (**b**) Fungi.

**Figure 7 microorganisms-12-01987-f007:**
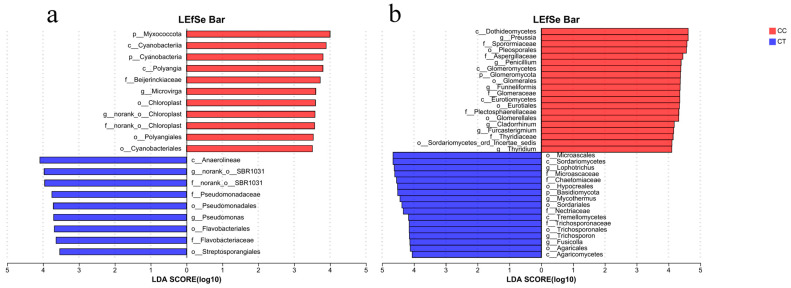
LDA discrimination results under CC and CT processing. (**a**) Bacteria. (**b**) Fungi.

**Figure 8 microorganisms-12-01987-f008:**
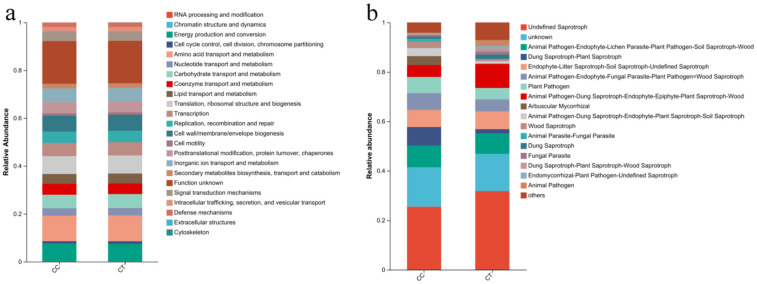
Function prediction of soil microorganisms under CC and CT treatments. (**a**) Bacteria. (**b**) Fungi.

**Table 1 microorganisms-12-01987-t001:** Effects of *T. harzianum* on growth indexes of *L. siceraria*.

Sample	Plant Height (cm)	Fresh Weight (g)	Dry Weight (g)
CC	18.06 ± 0.94 ^b^	9.02 ± 0.06 ^b^	3.29 ± 0.05 ^b^
CT	21.93 ± 1.70 ^a^	11.23 ± 0.15 ^a^	3.44 ± 0.03 ^a^

Note: Data are presented as mean ± SD. Different lowercase superscript letters indicate notable differences at *p* < 0.05.

**Table 2 microorganisms-12-01987-t002:** Effects of *T. harzianum* on soil chemical properties of *L. siceraria*.

Sample	SOC/(g/kg)	AP/(mg/kg)	AK/(mg/kg)	TN/(mg/kg)	EC/(mS/cm)	pH
CC	7.3 ± 0.34 ^b^	41.29 ± 1.52 ^b^	110.4 ± 1.89 ^b^	640.49 ± 12.27 ^b^	0.43 ± 0.01 ^b^	7.78 ± 0.11 ^a^
CT	9.3 ± 0.20 ^a^	50.60 ± 0.89 ^a^	134.0 ± 0.81 ^a^	745.63 ± 11.34 ^a^	0.49 ± 0.01 ^a^	7.51 ± 0.01 ^b^

Note: Data are presented as mean ± SD. Different lowercase superscript letters indicate notable differences at *p* < 0.05. SOC: soil organic carbon, AP: available phosphorus, AK: available potassium, TN: total nitrogen, and EC: electrical conductivity.

**Table 3 microorganisms-12-01987-t003:** Effects of *T. harzianum* on soil enzyme activities.

Sample	Urease (mg/g/d)	NAP (mg/g/d)	CAT (mg/g/d)	Sucrase (mg/g/d)
CC	0.48 ± 0.01 ^a^	0.97 ± 0.01 ^a^	1.20 ± 0.04 ^b^	7.89 ± 0.05 ^b^
CT	0.49 ± 0.02 ^a^	1.04 ± 0.06 ^a^	1.42 ± 0.06 ^a^	12.74 ± 0.44 ^a^

Note: Data are presented as mean ± SD. Different lowercase superscript letters indicate notable differences at *p* < 0.05. NAP: neutral phosphatase. CAT: catalase.

**Table 4 microorganisms-12-01987-t004:** Diversity along with richness indices of the bacterial and fungal communities under the CC and CT soil treatments.

	Sample	OTUs	ACE	Chao	Shannon	Simpson	Coverage
Bacterial	CC	3337.00 ± 280.83 ^a^	4076.75 ± 413.76 ^a^	3919.7 ± 361.43 ^a^	7.01 ± 0.09 ^a^	0.0022 ± 0.0001 ^a^	0.968223
	CT	3499.67 ± 63.01 ^a^	4278.05 ± 88.28 ^a^	4122.7 ± 106.67 ^a^	7.10 ± 0.05 ^a^	0.0021 ± 0.0002 ^a^	0.969683
Fungal	CC	499.67 ± 50.02 ^a^	512.96 ± 57.98 ^a^	516.86 ± 65.16^a^	4.36 ± 0.06 ^b^	0.0309 ± 0.0022 ^a^	0.999498
	CT	503.00 ± 11.53 ^a^	509.37 ± 10.85 ^a^	510.12 ± 11.04 ^a^	4.65 ± 0.10 ^a^	0.0215 ± 0.0022 ^b^	0.999693

Note: Data are presented as mean ± SD. Different lowercase superscript letters indicate notable differences at *p* < 0.05, Cutoff < 0.03.

## Data Availability

The original contributions presented in the study are included in the article, further inquiries can be directed to the corresponding author.

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
