# Peer review of "Effects of Trichoderma harzianum on Growth and Rhizosphere Microbial Community of Continuous Cropping Lagenaria siceraria"

_microorganisms, 2024, doi:10.3390/microorganisms12101987_

Round 1
Reviewer 1 Report
Comments and Suggestions for Authors
Always add line numbers to point out the errors in the manuscript easily
Abstract - "high-throughput sequencing technology" Which sequencing technology was adopted?
Abstract - "T. harzianum fertilizer may promote" Are you doubting your study? It means T harzianum may and may not contribute to the growth of...
"significantly. T. harzianum fertilizer may promote plant growth by both increasing the proportion of beneficial bacteria and reducing the proportion of harmful fungi on the one hand, and improving the physical and chemical properties of rhizosphere soil as well as enhancing soil enzyme activity. thereby mitigating the adverse effects of continuous crop-ping on L. siceraria." This statement should not appear in the abstract section. However, you could add a statement revealing the contribution to the knowledge of the research
Arrange the keywords alphabetically
Introduction: You could buttress your point on how Trichoderma fertilizer contributes to beneficial microbes inhabiting the rhizosphere soil. However, the resident microbial communities have been reported to promote plant growth promotion, be disease-resistant, and prevent abiotic stress and other challenges. What is the effectiveness of introducing Trichoderma fertilizer again?
Introduction: "Biotic and abiotic stress" The topic, abstract section, and content of the manuscript did not address this. Check
Methodology
Is this a green house or field experiment
what treatment did you perform on the collected soil samples before your plantation? I believe you are required to autoclave the soil to eliminate all microbes that may affect your experiment before introducing the sort of phytopathogens you would like to control and beneficial organisms that will be enhanced by the biofertilizer to promote their growth. The soil may also have a history of chemical or other biofertilizer application which may lead to error in your result.
Ref 17-21: These are secondary references.. Use Primary reference
Use high-resolution images and increase the size of the figures

The quality of the language is good but it requires English editing
Author Response
Comments 1: Always add line numbers to point out the errors in the manuscript easily. |
Response 1: We appreciate the reviewer’s comments. We added the line number to the full text. 2: Abstract-"high-throughput sequencing technology "Which sequencing technology was adopted?. Response 2: We appreciate the reviewer’s comments. We used the illumina miseq high-throughput sequencing technology and improved it in this paper. 3: "significantly. T. harzianum fertilizer may promote plant growth by both increasing the proportion of beneficial bacteria and reducing the proportion of harmful fungi on the one hand, and improving the physical and chemical properties of rhizosphere soil as well as enhancing soil enzyme activity. thereby mitigating the adverse effects of continuous crop-ping on L. siceraria." This statement should not appear in the abstract section. However, you could add a statement revealing the contribution to the knowledge of the research. Response 3: We appreciate the reviewer’s comments. Following the reviewer’s comments, we have revised the manuscript in lines 20-23. the contents have been revised as: “This study clarified that T. harzianum fertilizer promoted the growth of continuously planted L. siceraria plants by improving soil physical and chemical properties and microbial community structure, and was an effective microbial agent to alleviate the adverse effects of continuous cropping of L. siceraria” 4: Arrange the keywords alphabetically. Response 4: We appreciate the reviewer’s comments. Following the reviewer’s comments, we have revised the manuscript in lines 24-25. the contents have been revised as: “Keywords: Continuous cropping; Lagenaria siceraria; Microbial community; Trichoderma harzianum fertilizer” 5: Introduction: You could buttress your point on how Trichoderma fertilizer contributes to beneficial microbes inhabiting the rhizosphere soil. However, the resident microbial communities have been reported to promote plant growth promotion, be disease-resistant, and prevent abiotic stress and other challenges. What is the effectiveness of introducing Trichoderma fertilizer again? Response 5: We appreciate the reviewer’s comments. Following the reviewer’s comments, we have supplemented this part by consulting the literature. After the introduction of Trichoderma fertilizer, the relative abundance of beneficial microorganisms in the soil rhizosphere was increased. We have revised in line 92-97 of the manuscript. the contents have been revised as: “Under sterile laboratory conditions, the application of Trichoderma harzianum T-22 biofertilizer to carrot rhizosphere soil significantly increased the numbers of Bacillus and Pseudomonas, as well as beneficial fungi such as Trichoderma.” “Additionally, inoculation of T. harzianum ST02 in the rhizosphere soil of sweet sorghum significantly increased the relative abundance of beneficial nitrogen-fixing bacteria, such as Arthrobacter.” 6: Introduction: "Biotic and abiotic stress" The topic, abstract section, and content of the manuscript did not address this. Check. Response 6: We appreciate the reviewer’s comments. After checking the content, we have corrected it here, and have revised in lines 98-100 of the manuscript. the contents have been revised as: “Therefore, we investigated the effects of T. harzianum fertilizer on the performance of L. siceraria, as well as on the physico-chemical properties and microbial community structure of rhizosphere soil under continuous planting conditions.” 7: Is this a green house or field experiment? Response 7: We appreciate the reviewer’s comments. The experiment was carried out in the green house, we have revised in line 159 of the manuscript. 8: what treatment did you perform on the collected soil samples before your plantation? Response 8: We appreciate the reviewer’s comments. We refer to the treatment methods of soil in Liu, X. et.al (2023), Zhang, Z. et.al (2023) and Yu, T. et.al (2024) and other research literatures, and did not sterilize the soil. In this study, we used Trichoderma harzianum fertilizer, only added exogenous microbial agents without additional nutrition, and wanted to clarify the relationship between microbial agents, rhizosphere microorganisms, and hosts under the condition of continuous cultivation of L. siceraria. At the same time, we improved the detection of soil related properties and the environmental conditions of plant growth before the cultivation of L. siceraria. We have added the manuscript in lines 113-153. References are as follows: Liu, X.; Ren, X.; Tang, S.; Zhang, Z.; Huang, Y.; Sun, Y.; Gao, Z.; Ma, Z. Effects of Broccoli Rotation on Soil Microbial Community Structure and Physicochemical Properties in Continuous Melon Cropping. Agronomy 2023, 13, 2066. https://doi.org/10.3390/agronomy13082066 Zhang, Z.; Tang, S.; Liu, X.; Ren, X.; Wang, S.; Gao, Z. The Effects of Trichoderma viride T23 on Rhizosphere Soil Microbial Communities and the Metabolomics of Muskmelon under Continuous Cropping. Agronomy 2023, 13, 1092. https://doi.org/10.3390/agronomy13041092 Yu, T., Sun, Q., Liu, Z. et al. The Application of Orychophragmus violaceus as a Green Manure Relieves Continuous Cropping Obstacles in Peanut Cultivation by Altering the Soil Microbial Community and Functional Gene Abundance. J Soil Sci Plant Nutr (2024). https://doi.org/10.1007/s42729-024-01867-x 9: Ref 17-21: These are secondary references. Use Primary reference? Response 9: We appreciate the reviewer’s comments. Following the reviewer’s comments, we have revised in lines 177-183 of the manuscript. 10: Use high-resolution images and increase the size of the figures Response 10: We appreciate the reviewer’s comments. Following the reviewer’s comments, we put all the pictures in the content, and improved the resolution, increased the image size. In addition, we also checked and corrected the errors marked in the original text. |

Reviewer 2 Report
Comments and Suggestions for Authors
Dear Authors
Your manuscript is interesting because it presents effects of Trichoderma harzianum fertilizer on growth and rhizosphere microbial community of continuous cropping Lagenaria siceraria. The layout of the articles became standard for research papers. I don't feel qualified to judge about the English language and style. However, I have a few comments:
General comments:
The title and abstract correspond to the content of the manuscript.
The introduction contains the most important information about continuous cropping Lagenaria siceraria and effects of Trichoderma harzianum fertilizer. I am a soil scientist and I study soil biology, so the introduction lacks more information about the impact of gourd cultivation on soils, and information about soil enzymes. However, this is just my opinion. It is not necessary to include this information.
Research objective. Is the use of biocontrol agents in gourd cultivation new? Are your studies pilot? Write what is innovative in your research.
The research methods were well characterized. However, there is no characterization of the soil tested before gourd planting. There is also no description of the conditions in which plant production took place. This needs to be completed.
The results, discussion and conclusion were well described. However, there is no recommendation regarding the use of this bio-agent.
Minor faults are listed below.
Detailed comments
Check the entire text and insert a space between the word and the square bracket. à “herb[1]“
Abstract: "The pH of rhizosphere soil decreased from 7.78 to 7.51, while the available phosphorus (AP), the electrical conductivity (EC), the available potassium-sium (AK), and the total nitrogen (TN)". This sentence is a mess. Group the results by category. pH, EC, AP, AK.
Results:
Table 3. The table is incomplete.
Figure 6. The drawing is illegible. Please improve its quality.
Figure 7. The drawing is illegible. Please improve its quality.
Figure 8. The drawing is illegible. Please improve its quality.
Discussion: In the discussion you refer to different biotypes of Trichoderma. Are you able to specify which biotype you used?
Section 4.3. “[38]. In addition, Many strains” correct this part of the sentence.
Missing section name "references".
Please prepare references according to the guidelines given in the instructions for authors (https://www.mdpi.com/journal/plants/instructions).
Good luck!
Sincerely yours
Reviewer
Author Response
Comments 1: The introduction contains the most important information about continuous cropping Lagenaria siceraria and effects of Trichoderma harzianum fertilizer. I am a soil scientist and I study soil biology, so the introduction lacks more information about the impact of gourd cultivation on soils, and information about soil enzymes. |
Response 1: We appreciate the reviewer’s comments. At present, although it has been reported in cucurbitaceae (watermelon, cucumber, etc.) plants, there are few articles on soil information after continuous cropping of L. siceraria. Other projects have determined the effect of L. siceraria cultivation on soil and are also being submitted. We hope the reviewer can understand. 2: Research objective. Is the use of biocontrol agents in gourd cultivation new? Are your studies pilot? Write what is innovative in your research. Response 2: We appreciate the reviewer’s comments. In this study, Trichoderma harzianum fertilizer was used to treat continuous planting L. siceraria for the first time. By comparing with the continuous cropping group, it was further confirmed that the T. harzianum fertilizer could promote the growth of continuous cropping gourd. Studies have shown that on the one hand, it can promote the increase of plants, as well as the increase of dry weight and fresh weight. On the other hand, it can improve the physical and chemical properties of rhizosphere soil and the activity of soil enzymes, and improve the nutrient content and conversion efficiency in soil. Most importantly, for the first time, from the perspective of rhizosphere microbial community, through high-throughput sequencing, it was found that T. harzianum fertilizer could increase the relative abundance of soil beneficial bacteria and reduce the relative abundance of harmful bacteria, revealing the mechanism of alleviating the adverse effects of L. siceraria continuous cropping. 3: The research methods were well characterized. However, there is no characterization of the soil tested before gourd planting. There is also no description of the conditions in which plant production took place. This needs to be completed. Response 3: We appreciate the reviewer’s comments. Following the reviewer’s comments, we have revised the manuscript in lines 113-153. “The soil used for the pot experiment was sourced from the plantation of the College of Agronomy and Biology at Liaocheng University, located in Liaocheng City, Shandong Province, China, where L. siceraria has been continuously cultivated for two years. This region falls within the temperate monsoon climate zone, with an annual average temperature of approximately 13.1 °C, annual precipitation of about 600 mm, and annual frost-free period of about 200 days. Before planting, the soil had a pH of 7.23, organic carbon content of 8.20 g/kg, total nitrogen of 710.45 mg/kg, available phosphorus of 45.63 mg/kg, available potassium of 121.52 mg/kg, and electrical con-ductivity (EC) of 0.56 mS/cm.” 4: The results, discussion and conclusion were well described. However, there is no recommendation regarding the use of this bio-agent. Response 4: We appreciate the reviewer’s comments. Following the reviewer’s comments, we identified the type of Trichoderma harzianum fertilizer (Trichoderma harzianum T-22). After consulting the literature, we explained the specific use of T. harzianum biological agents. The methods of its use are diverse, including root irrigation, soil mixing, spraying, etc. They have promoted the growth of plants. In this experiment, the method of root irrigation was adopted. According to the instructions provided by the merchants, the effect observed was also very good. And we have revised the manuscript in lines 421-434. 5: Check the entire text and insert a space between the word and the square bracket. À “herb [1]. Response 5: We appreciate the reviewer’s comments. Following the reviewer’s comments, I have checked and revised the full text. 6: Abstract: "The pH of rhizosphere soil decreased from 7.78 to 7.51, while the available phosphorus (AP), the electrical conductivity (EC), the available potassium-sium (AK), and the total nitrogen (TN)". This sentence is a mess. Group the results by category. pH, EC, AP, AK. Response 6: We appreciate the reviewer’s comments. Following the reviewer’s comments, we have revised in lines 13-15 of the manuscript. 7: Minor faults of results: Response 7: We appreciate the reviewer’s comments. We improved Table 3 and added ' sample ' in Tables 1, 2 and 3. In addition, we have revised all of the above issues. |
